# Transgenerational effects of early life stress on the fecal microbiota in mice
Nize Otaru[1,2], Lola Kourouma[3,4], Benoit Pugin[2], Florentin Constancias[2], Christian Braegger[1], Isabelle M. Mansuy ⑩ [3,4,5] ✉ & Christophe Lacroix ⑩ [2,5] ✉

Stress in early life can affect the progeny and increase the risk to develop psychiatric and cardiometabolic diseases across generations. The cross-generational effects of early life stress have been modeled in mice and demonstrated to be associated with epigenetic factors in the germline. While stress is known to affect gut microbial features, whether its effects can persist across life and be passed to the progeny is not well defined. Here we show that early postnatal stress in mice shifts the fecal microbial composition (binary Jaccard index) throughout life, including abundance of eight amplicon sequencing variants (ASVs). Further effects on fecal microbial composition, structure (weighted Jaccard index), and abundance of 16 ASVs are detected in the progeny across two generations. These effects are not accompanied by changes in bacterial metabolites in any generation. These results suggest that changes in the fecal microbial community induced by early life traumatic stress can be perpetuated from exposed parent to the offspring.

It has long been established, that early life is a critical developmental window in which individuals are primed for life[1,2]. Traumatic stress in early life is one of the major causes of mental and physical diseases, particularly psychiatric disorders. These diseases often not only affect individuals directly exposed to early life stress but also sometimes their offspring[3–5]. Thus, accumulating evidence indicates that the effects of early life stress can be manifested by subsequent generations, potentially affecting their behavior and physiology[6]. A first study by Weaver et al. (2004) showed that maternal care in rats may alter the stress responses of the offspring by inducing epigenetic changes, particularly changes in DNA methylation at the promoter of the glucocorticoid receptor gene in the hippocampus. This effect is mediated by maternal care and needs exposure at each generation[7]. Transmission of the effects of early life stress involving the germline was however demonstrated in mice. Male mice subjected to early life stress have behavioral and metabolic symptoms that can be manifested in their offspring until the fourth or even fifth generation[8,9]. Evidence of cross-generation effects of early life stress in humans is scarcer and remains correlative[10] but suggest a link between adversity and gut microbiome composition[11].

Alterations in signaling pathways *via* the gut-brain axis have been suggested to contribute to the consequences of stress exposure[12]. Signaling *via* gut-brain axis is thought to occur by bacterially produced metabolites,

such as short chain fatty acids (SCFAs) and γ-aminobutyric acid[13,14]. Since gastrointestinal tract maturation and the establishment of the gut microbiota occur in parallel with brain development[15], alterations in the gut microbiome may contribute to brain dysfunctions later in life. Experiments with germ free mice have indeed suggested the involvement of gut microbes in anxiety like behavior and behavioral despair induced by early life stress[16]. Early life stress has been shown to induce distinct gut microbial alterations in rodent models including the decrease of *Lactobacillaceae* species and increase in *Enterococcus* species. Though, observed changes are heterogenous and differ substantially between studies[17,18]. Similar observations exist in humans, though observed stress effects on the gut microbiota remain correlative and cannot be distinguished from confounding effects such as diet and lifestyle[17,19,20].

Previous research has mainly focused on changes in gut microbial composition rather than changes in function. In the few available human studies referring to gut microbial metabolic changes associated with early life stress, socioeconomic risk exposure (e.g., low income to needs ratio and parent-child dysfunction) explained a significant amount of the functional gut microbial diversity[21] and a decrease of four gut metabolites (i.e., glutamate gamma-methyl ester, 5-oxoproline, malate, and urate) was observed when comparing individuals with low and high early life adversity

[1]Nutrition Research Unit, University Children's Hospital Zürich, Zürich, Switzerland. [2]Department of Health Sciences and Technology, Laboratory of Food Biotechnology, ETH Zürich, Zürich, Switzerland. [3]Department of Health Science and Technology of the ETH Zurich, Laboratory of Neuroepigenetics, Brain Research Institute, Medical Faculty of the University of Zurich, and Institute for Neuroscience, Zurich, Switzerland. [4]Center for Neuroscience Zürich, ETH and University Zürich, Zurich, Switzerland. [5]These authors jointly supervised this work: Isabelle M. Mansuy, Christophe Lacroix. ✉e-mail: mansuy@hifo.uzh.ch; christophe.lacroix@hest.ethz.ch

exposure[19]. Given the effects of early life stress on brain functions and health in the progeny, it is conceivable that early life stress induced alterations in the gut microbial community can also affect the offspring. Today, a potential intergenerational effect of early life stress on the gut microbiota has not been carefully examined.

The mouse model of unpredictable maternal separation combined with unpredictable maternal stress (MSUS) distinguishes itself from other rodent models of stress by applying unpredictability as traumatic factor and by combining stress of mothers and of pups (daily unpredictable separation for 3 h and in addition mothers are stressed unpredictably during separation by restraint or forced swim)[22]. This model has been extensively characterized and shown to induce behavioral, metabolic and physiological alterations across generations[9,23]. Epigenetic factors in the germline have been proposed as mediators of transmission and expression of phenotypes, and sperm RNA was causally demonstrated to be a vehicle of transmission between exposed father and offspring and grand-offspring [22,24,25]. Although, the MSUS paradigm has been shown to induce visceral sensitivity [26], the effects of early life stress on the gut microbial community across generations has not been evaluated before. Here, we show that postnatal traumatic stress modifies the fecal microbial community in directly exposed mice but also in their offspring.

## Results
### Dynamics of fecal microbiota across life in healthy mice
To determine the dynamics of the fecal microbial community and its functionality across life in mice, the fecal microbiota from late postnatal to adult stages was analyzed.

Fecal microbial alpha diversity – the diversity within a community – was determined by assessing the richness and evenness of the community. Richness was defined as observed amplicon sequencing variants (ASVs) in a sample. Evenness was assessed with the Pielou's index, a measure indicating how evenly ASVs are distributed in a sample. Shannon diversity combines both richness and evenness in one function. After weaning, microbial richness rapidly increased from median values of 226 (iqr: 54) to 348 (iqr: 57) observed ASVs in 22-day-old mice and 30-day-old mice, respectively (Fig. 1). Richness continued to increase across life span, although at a lower rate ($\log_2$ fold-change), and reached a plateau in 15-week-old mice (Fig. 1). In contrast, Pielou's and Shannon's index were the highest in 30-day-old mice but did not change thereafter (Fig. 1).

Beta diversity – diversity between communities – was evaluated using qualitative (binary Jaccard index) and quantitative (weighted Jaccard index) metrics. Fecal microbial composition was investigated *via* binary Jaccard index, and microbial structure was investigated *via* weighted Jaccard index. Both, microbial composition and structure rapidly changed after weaning ($p < 0.01$ FDR-adjusted; $R^2 = 0.162$ and $R^2 = 0.180$) from 22-day-old to 30-day-old mice (Fig. 2). Significant ($p < 0.05$, FDR-adjusted) differences in microbial composition and structure were observed throughout life span, with lower effect size (determination coefficient; $R^2$) with increasing age (Fig. 2). In addition, 402 uniquely differently abundant ASVs ($p < 0.05$, FDR-adjusted) were detected across life (Supplementary Fig. 1). In concurrence with changes observed in beta diversity, most differently abundant ASVs were observed comparing 22-day-old to older mice. Approximately half (176 ASVs) of differently abundant ASVs belonged to the family *Lachnospiraceae* (Supplementary Fig. 1).

To evaluate the effect of the temporal succession of behavioral testing and breeding, fecal microbiota of 30-week-old mice was examined in 2 different conditions: behavioral phenotyping conducted before breeding (GR1) or breeding conducted before behavioral testing (GR2). No significant differences in alpha diversity (richness, evenness, and Shannon's diversity) and beta diversity (binary and weighted Jaccard index) were observed between groups (Supplementary Table 1). However, 15 differentially abundant ASVs ($p < 0.05$, FDR-adjusted) were identified, all different in the range of 1.3 to 3.4 $\log_2$ fold-changes (Fig. 3). Relative abundance of these ASVs was generally low, with individual

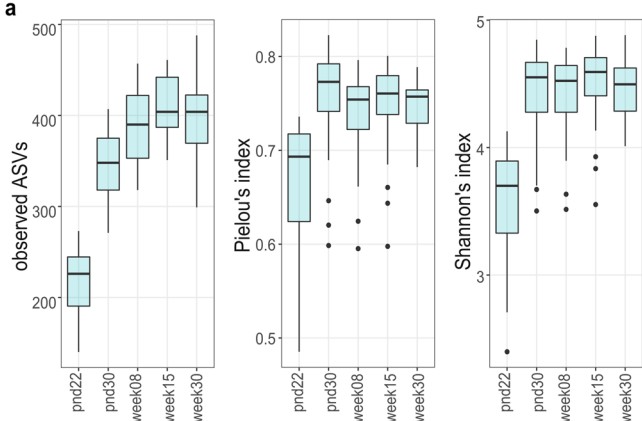

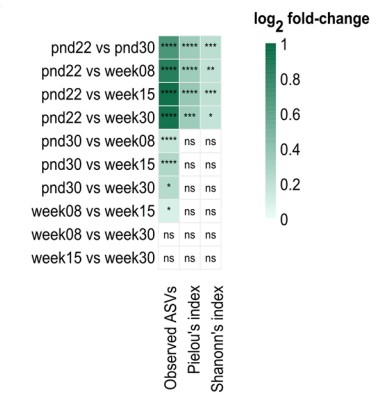

**Fig. 1 | Effect of age on alpha diversity metrics of control mice fecal microbiota. a** Fecal microbial richness (observed ASVs), evenness (Pielou's index), and Shannon-diversity across life. Boxplot with box elements showing upper and lower quantile and median. Whiskers extend from the hinge to ±1.5 times the interquartile range or the highest/lowest value. Outliers are indicated as black points. **b** $\log_2$ fold-change in alpha diversity metrics per pairwise comparison of different ages. Significance was calculated using $\log_2$ transformed metrics and generalized mixed effect models with FDR correction. Number of animals per age are stated in Supplementary Table 4. ns: not significant; *$p < 0.05$; **$p < 0.01$; ***$p < 0.001$; ****$p < 0.000$.

microbiota values ranging from 0% to 7.3% (Supplementary Fig. 2). In contrast, no differences in measured cecal bacterial organic acids, amino acids and amines were detected (Supplementary Data 1; Supplementary Figs. 3, 4).

### Early life stress induces a shift in fecal microbial community in directly exposed mice across life
To investigate if early life stress (MSUS paradigm) has a long-lasting effect on the fecal microbial community across life, the fecal microbiota was compared between F1 MSUS and control mice across the life span from 22 days till 30 weeks of age. In addition, animal weight was monitored in mice from 8 weeks to 30 weeks of age. Development of mice weight was not different between MSUS and controls ($p > 0.05$, FDR-adjusted) at different ages (Supplementary Fig. 5).

Different alpha diversity metrics were compared between F1 MSUS and control groups at different ages. No significant differences ($p > 0.05$, FDR-adjusted) in fecal microbial richness, evenness, and Shannon's diversity index between MSUS and controls were observed, with comparable median values for both groups at different ages (Fig. 4).

Microbiota composition significantly differed between MSUS and controls in 30-day-old, 8-week-old, 15-week-old, and 30-week-old mice (binary Jaccard index; $p < 0.01$, FDR-adjusted; Fig. 5). Microbiota structure significantly differed between MSUS and controls in 30-day-old mice

**Fig. 2 | Effect of age on beta diversity metrics of control mice fecal microbiota.** Visualization as principal correspondence analysis of (**a**) binary (microbial composition) and (**b**) weighted (microbial structure) Jaccard index. Individual symbols display aggregated microbiota per litter (pnd22) or cage (pnd30, week08, week15, and week30), large symbols display centroids, ellipses indicate 95% of confidence intervals. **c** Heatmap of $R^2$ for pairwise comparison of different ages. Significance was calculated using PERMANOVA including FDR correction. Number of animals per age are stated in Supplementary Table 4. Pairwise comparison of different dispersions between ages are listed in Supplementary Table 2. pnd: postnatal day; *$p < 0.05$ ;**$p < 0.01$.

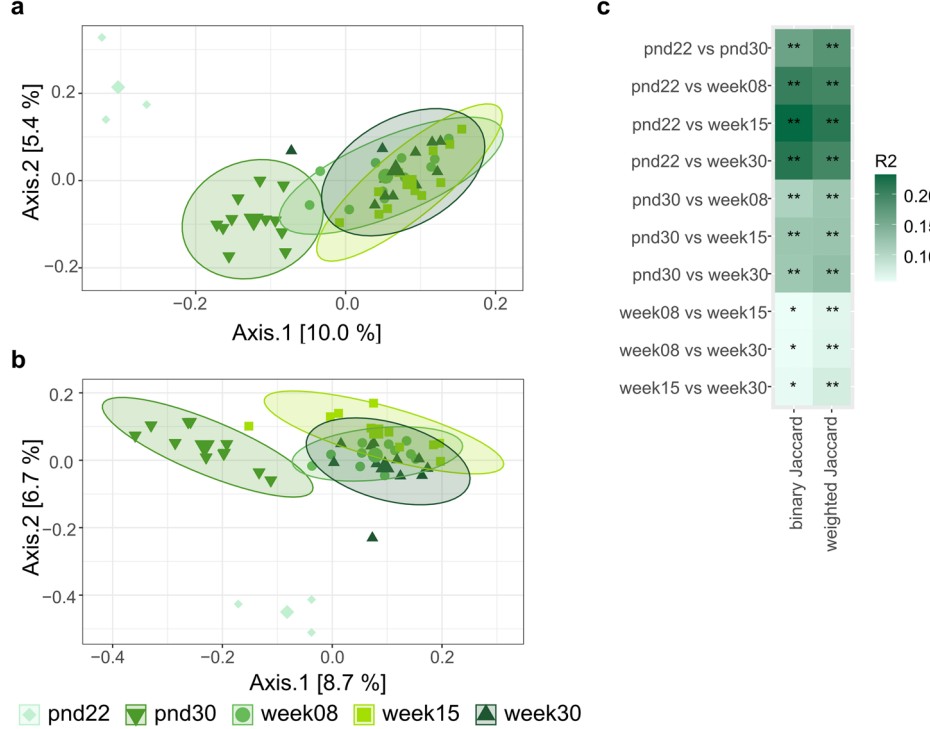

**Fig. 3 | Effect of temporal succession of behavioral phenotyping and breeding on 30-week-old control mice fecal microbiota.** Comparison of phenotyping group 2 (GR2; breeding before behavioral phenotyping; $n = 15$) *versus* group 1 (GR1; behavioral phenotyping before breeding; $n = 16$) microbiota. Log$_2$ fold-change in specific ASVs significantly ($p < 0.05$, FDR-adjusted) decreased or increased in GR2 compared to GR1. Taxonomic information is indicated at family level. Relative abundances of individual ASVs are depicted in Supplementary Fig. 2. Significance was calculated using log$_2$ transformed abundance counts and generalized mixed effect models with FDR correction.

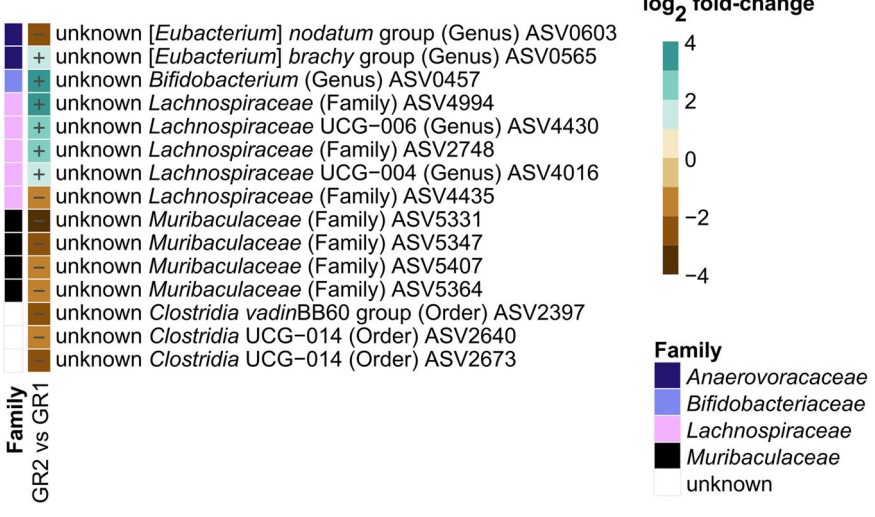

(weighted Jaccard index; $p < 0.01$, FDR-adjusted; Fig. 5). Overall effect size was generally low with determination coefficients ($R^2$) ranging from 0.058 to 0.067 (Fig. 5; Supplementary Table 3). Contrary to beta diversity, differentially abundant ASVs ($p < 0.05$, FDR-adjusted) were only detected for 8-week-old and 15-week-old mice, in the range of 0.98 to 3.06 log$_2$ fold-changes (Fig. 6). Yet, overall abundance of these ASVs was low, with individual microbiota values ranging from 0% to 1.9% (Supplementary Figs. 6, 7). One ASV belonging to the Family *Ruminococcaceae* (ASV1948) was persistently decreased in MSUS compared to controls in both 8-week-old and 15-week-old mice.

As a read out for the bacterial metabolic potential, cecal bacterial organic acids, amino acids, and amines were evaluated for 30-week-old mice. No significant ($p > 0.05$, FDR-adjusted) differences between MSUS and controls were observed, with comparable median values for both groups (Supplementary Figs. 8, 9).

## Early life stress shifts the trajectory of fecal microbial community across two generations

To investigate if early life stress in F1 alters the fecal microbial community in the progeny, fecal microbiota of mature adult MSUS and control mice (aged 28 or 30 weeks) were compared in the patriline (F2 and F3). It must be noted, that F3 mice were not direct offspring of investigated F2 mice, but of a subset of F2 mice which were not included in this study (Fig. 7). Akin to results observed for F1 mice across life (Supplementary Fig. 5), development of mice in terms of animal weight was consistent between MSUS and control mice in F2 and F3 (Supplementary Fig. 10).

To characterize the fecal microbial community, alpha diversity, beta diversity and differentially abundant ASVs were evaluated for each generation. Similar to results observed in F1 across life, fecal microbial richness, evenness, and Shannon's diversity did not significantly, differ between MSUS and controls for F2 and F3 (Supplementary Fig. 11). As F1, F2, and F3

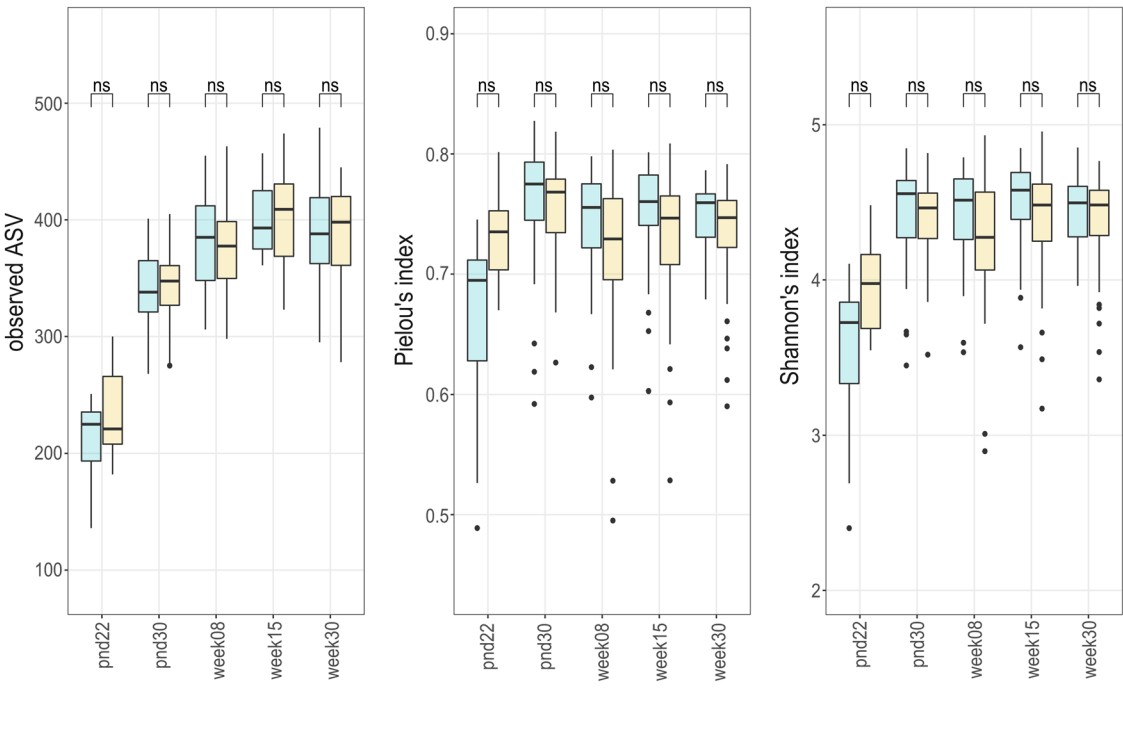

**Fig. 4 | Comparison of different alpha diversity metrics between F1 MSUS and control fecal microbiota across life span.** Comparison of richness (observed ASV), evenness (Pielou's index), and Shannon-diversity between MSUS and controls per time point. Boxplot with box elements showing upper and lower quantile and median. Whiskers extend from the hinge to ±1.5 times the interquartile range or the highest/lowest value. Outliers are indicated as black points. Significance was calculated using $\log_2$ transformed metrics and generalized mixed effect models with FDR correction. Number of animals per group are stated in Supplementary Table 4. pnd: postnatal day; ns: not significant.

were sequenced in different runs (confounding factor), alpha diversity between generations was not directly compared.

Contrary to observations in mature adult F1 mice, where only an effect of the MSUS paradigm on microbiota composition (binary Jaccard index) was observed, both fecal microbial composition and structure significantly differed (weighted Jaccard index; $p < 0.05$, FDR-adjusted) between MSUS and controls in the patriline (F2 and F3; Fig. 8). Effect size was highest in F3 with determination coefficients ($R^2$) of 0.130 and 0.099 for binary and weighted Jaccard index, respectively. This was approximately 2-fold higher when compared to F1 and F2 (Fig. 8; Supplementary Table 4). While no differently abundant ASVs were identified for mature adult F1 mice, in the patriline differently abundant ASVs ($p < 0.05$, FDR-adjusted), differed in the range of 2.1 to 5.7 $\log_2$ fold-changes (Fig. 9). Yet, overall abundance of these ASVs was low, with individual microbiota values ranging from 0% to 2.7% (Supplementary Figs. 12, 13). Half of observed uniquely differentially abundant ASVs, belonged to the family *Lachnospiraceae*. One ASV, belonging to the family *Muribaculaceae* (ASV5777), was persistently decreased in MSUS compared to controls in both F2 and F3 (Fig. 9).

To investigate, if observed differences in fecal microbial community result in function changes, bacterial metabolites were quantified in cecal content. Akin to observations in F1, no significant differences in cecal bacterial organic acid, amino acid, and amine concentrations were detected in 28-week-old mice of F2 (Supplementary Figs. 14, 15). Cecal bacterial metabolites were not evaluated in F3.

**Fecal microbial community in offspring is not directly influenced by the father microbiota**

To investigate, if observed differences in F2 MSUS *versus* controls stem from a transmission of fecal microbiota of MSUS-fathers to mothers and subsequently to offspring, beta diversity distances between pups and their parents, and between mating and non-mating pairs was evaluated. To minimize a potential effect of different phenotyping groups, father microbiota was evaluated before breeding or phenotyping, when mice were 16 weeks old.

Fecal microbial composition of 21-day-old pups was significantly (binary Jaccard index; $p < 0.0001$; FDR-adjusted) closer to mother microbiota than father microbiota for both MSUS and controls. Fecal microbial structure (weighted Jaccard index) was either as close to mother and father microbiota or significantly ($p < 0.05$; FDR-adjusted) closer to mother microbiota, for MSUS and control pups, respectively (Fig. 10a). The older the pups (from 22-days-old to 16-weeks-old), the more similar the fecal microbial composition and structure got to parent microbiota, displayed by overall lower distances between microbiota. Microbial composition of 16-week-old pups remained significantly ($p < 0.001$; FDR-adjusted) closer to mothers than fathers. Microbial structure was either significantly ($p < 0.01$; FDR-adjusted) closer to father than mother microbiota, or as close to father and mother microbiota, for MSUS or control pups, respectively (Fig. 10a). Mating and non-mating pairs showed no significant ($p > 0.05$; FDR-adjusted) difference in fecal microbial composition and structure. The temporal differences between sample collection and mating did not have an effect on observed distances, as indicated by absence of clustering according to phenotyping group (Fig. 10b).

## Discussion

Early life traumatic stress induces metabolic, and behavioral alterations across generations[9,23]. This study presents the first in-depth assessment of the effect of early life traumatic stress on the gut microbial community and metabolic functions in MSUS mice, while also addressing a transgenerational effect.

During the course of life distinct gut microbial changes occur, though in humans it is difficult to distinguish between lifestyle changes and the aging process itself[27,28]. In line with human studies[29], here we show that the

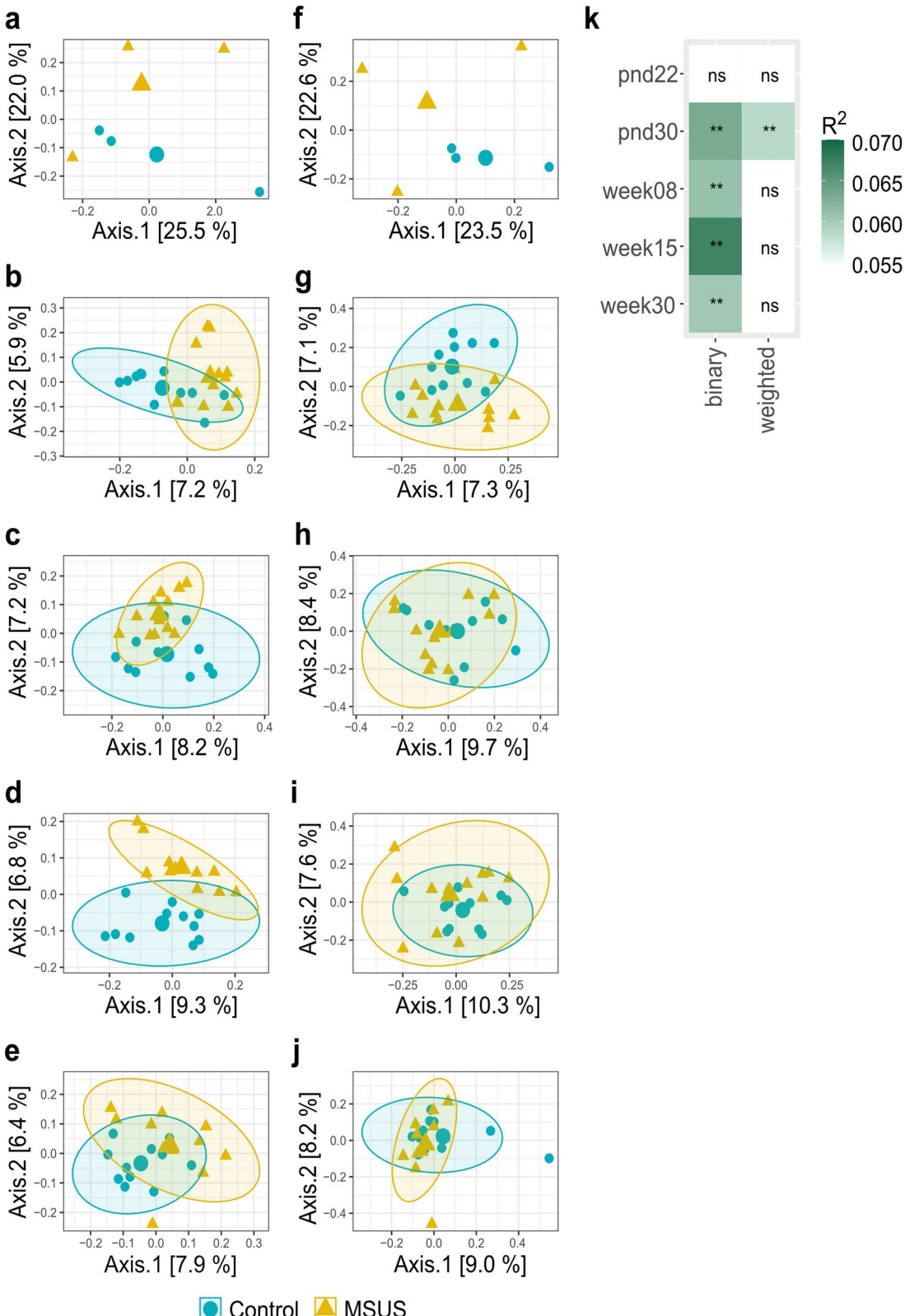

Control    MSUS

**Fig. 5 | Effect of the MSUS paradigm on beta diversity metrics of fecal microbiota across life.** Comparison of MSUS *versus* control mice is displayed. Visualization as principal correspondence analysis of (**a−e**) binary (microbial composition) and (**f−j**) weighted Jaccard index (microbial structure). **a, f** pnd22, (**b, g**) pnd30, (**c, h**) week08, (**d, i**) week15, and (**e, j**) week30 are depicted. Small symbols display aggregated microbiota per litter (pnd22) or cage (pnd30, week08, week15, and week30), large symbols display centroids, ellipses indicate 95% of confidence intervals. **k** Heatmap of $R^2$ for comparison between MSUS and control at different time points. Significance was calculated using PERMANOVA including FDR correction. Number of animals per group are stated in Supplementary Table 4. Comparisons of different dispersions between MSUS and controls at different time points are listed in Supplementary Table 3. pnd: postnatal day; ns: not significant; **$p < 0.01$.

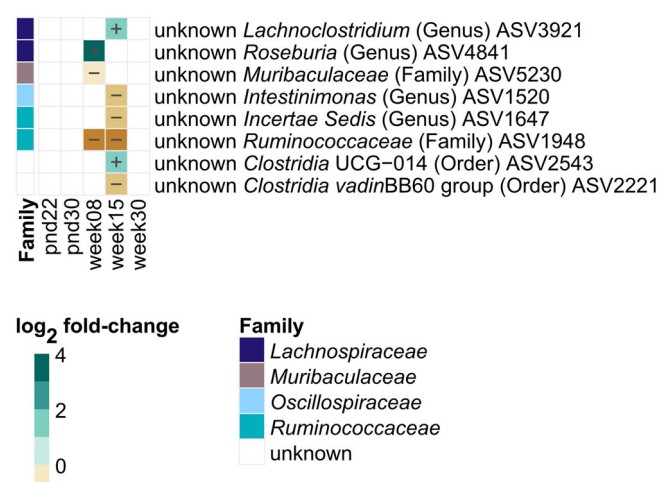

**Fig. 6 | Log2 fold-change in specific ASVs significantly (*p* < 0.05, FDR-adjusted) decreased or increased in F1 MSUS compared to controls over life.** Taxonomic information is indicated at family level. Relative abundances of individual ASVs are depicted in Supplementary Figs. 6, 7. Significance was calculated using log₂ transformed abundance counts and generalized mixed effect models with FDR correction. Number of animals per group are stated in Supplementary Table 4. pnd: postnatal day.

cessation of breastmilk has the most pronounced effect on the murine gut microbiota, illustrated by changes in gut microbial features including richness, evenness, structure, and composition. Although richness and evenness stay relatively stable in adulthood, significant changes are still observed over time with respect to structure and composition, suggesting a significant effect of the aging process itself. This is in line with previous studies addressing age-related changes in the murine gut microbiota, demonstrating the dynamics of the fecal microbial community and its metabolic functions from young adult (approximately 8 weeks of age) to old (older than 100 weeks of age) mice[30–32]. We show that the temporal succession of breeding and behavioral phenotyping of mice has a significant effect on relative abundances of several fecal microbial ASVs. Both acute and chronic stress in adulthood have been shown to alter the fecal/cecal microbial community and related metabolic functions[33–39]. As the applied behavior phenotyping methods, i.e., forced swim test, pose as an acute stressor themselves, it is conceivable that observed alterations originate in the temporal differences between acute stressor and fecal sample collection. However, fecal sample collection for both groups (behavioral phenotyping before breeding *versus* breeding before behavior phenotyping) was performed at least two months after behavioral phenotyping. Whether two months are indeed sufficient to alleviate acute stress effects on the gut microbiota remains to be explored.

Early life stress has previously been shown to alter the murine fecal microbial community in studies focusing on a single time point after exposure[18,40,41]. Here we show in a longitudinal study that early life traumatic stress in mice results in significant changes in fecal microbial community composition (binary Jaccard index) persistent across life. Significantly differentially abundant ASVs were largely different across life. ASVs belonging to the genus *Lachnoclostridium* were previously shown to be enriched in early life maternally separated mice[18], which we confirm here. We show that one ASV belonging to the family *Ruminococcaceae* is persistently decreased in both 8 and 15-week-old mice, in line with a previous study, illustrating the depletion of corresponding ASVs after early life stress exposure[18]. The *Ruminococcaceae* family includes prominent butyrate producers[42], suggesting a modulation of the SCFA-profile by early life stress. Yet, here we report no significant changes in

cecal SCFAs, amines, amino acids between early life stress exposed and non-exposed 30-week-old mice. In contrary, in the only available study where the gut bacterial metabolome in mice was investigated upon early life stress exposure, a decrease in fecal lactate an increase in cholic acid in early life stressed mice compared to controls was observed[43]. Some previous studies found significant decreases in alpha diversity metrics upon early life stress exposure, while no significant difference in fecal microbial alpha diversity between early life stress exposed and non-exposed rodents were detected in the present and other studies[18,41,44,45]. These substantial differences between studies may result from different stress paradigms applied (i.e., maternal separation and/or early weaning or limited bedding and nesting), or from mouse husbandry factors, such as cage ventilation, chow, and bedding, which have previously been shown to substantially affect the gut microbial community[46–48].

Previous studies have shown the change of gut microbial features upon exposure to acute stress (e.g., heat, light, cold water, and restraint stress) in adult rodents[33–36]. Thus, it is conceivable that the applied stresses mothers are exposed to during the MSUS paradigm (i.e., restraint and acute swim stress) induce similar changes in the gut microbial community. These changes may then affect offspring, as it has been shown that disruption of mother's gut microbiota by antibiotics during nursing altered fecal microbial community and adaptive immunity in offspring[49]. Nevertheless, Kemp and colleagues (2021) suggest that gut microbial alterations due to early life stress occur independent of maternal microbiota inheritance.

The present study shows a transgenerational effect of early life traumatic stress induced changes on the fecal microbial community. Significant differences in fecal microbial composition (binary Jaccard index), structure (weighted Jaccard index), and relative abundances of several ASVs, between MSUS and control mice up to F3 were observed. We also show an increase in effect size with increasing number of generations (F1 to F3). However, only one ASV belonging to the family *Muribaculaceae* was persistently decreased in MSUS compared to controls in F2 and F3, suggesting different shifts in fecal bacterial community in MSUS mice of different generations. Yet, different ASVs belonging to the family *Muribaculaceae* were depleted in both F2 and F3. The *Muribaculaceae* abundance has previously been correlated with T-cells[50] depressive-like[51], and autism-like- behavior[52]. In line with directly early life stress exposed mice[18,41,53], we show a modulation of *Lachnospiraceae* ASVs in both F2 and F3. Abundance of the family *Lachnospiraceae* has previously been negatively correlated with depressive-like behavior in mice[54]. The family *Lachnospiraceae* harbors major butyrate and propionate producers[42], while *Muribaculaceae* levels have previously been associated with propionate production[55]. This may imply a modulation of SCFA levels in F2 and F3, which we could not confirm here. This discrepancy could be explained by different factors, including absorption of SCFAs in the cecum, time of feeding before sacrifice, and the concept of functional redundancy[56,57]. Nevertheless, these observations suggest a crucial role of both *Muribaculaceae* and *Lachnospiraceae* species in the microbiota-host crosstalk. Similarly, experiments with germ free mice have suggested a crucial involvement of gut microbes in anxiety-like behavior and behavioral despair induced by early life stress[16]. After colonization of early life stressed germ-free mice distinct shifts in gut microbiota profiles were observed, which were not present in unstressed controls. It was only with said colonization that differences in anxiety-like behavior and behavioral despair between early life stressed mice and unstressed controls were detected[16]. These observations suggest that early life stress exposure modulates host factors (e.g., immune tone[58,59]). These host factors may be responsible for modulation of gut microbes. The altered gut microbial community may then signal back to the host exhibiting phenotypic changes. This notion could be present in F2 and F3 MSUS mice explaining observed gut microbial changes without directly experiencing early life traumatic stress beforehand.

Though, the present study exhibits various strength, including relatively large sample size, monitoring of fecal microbial changes over

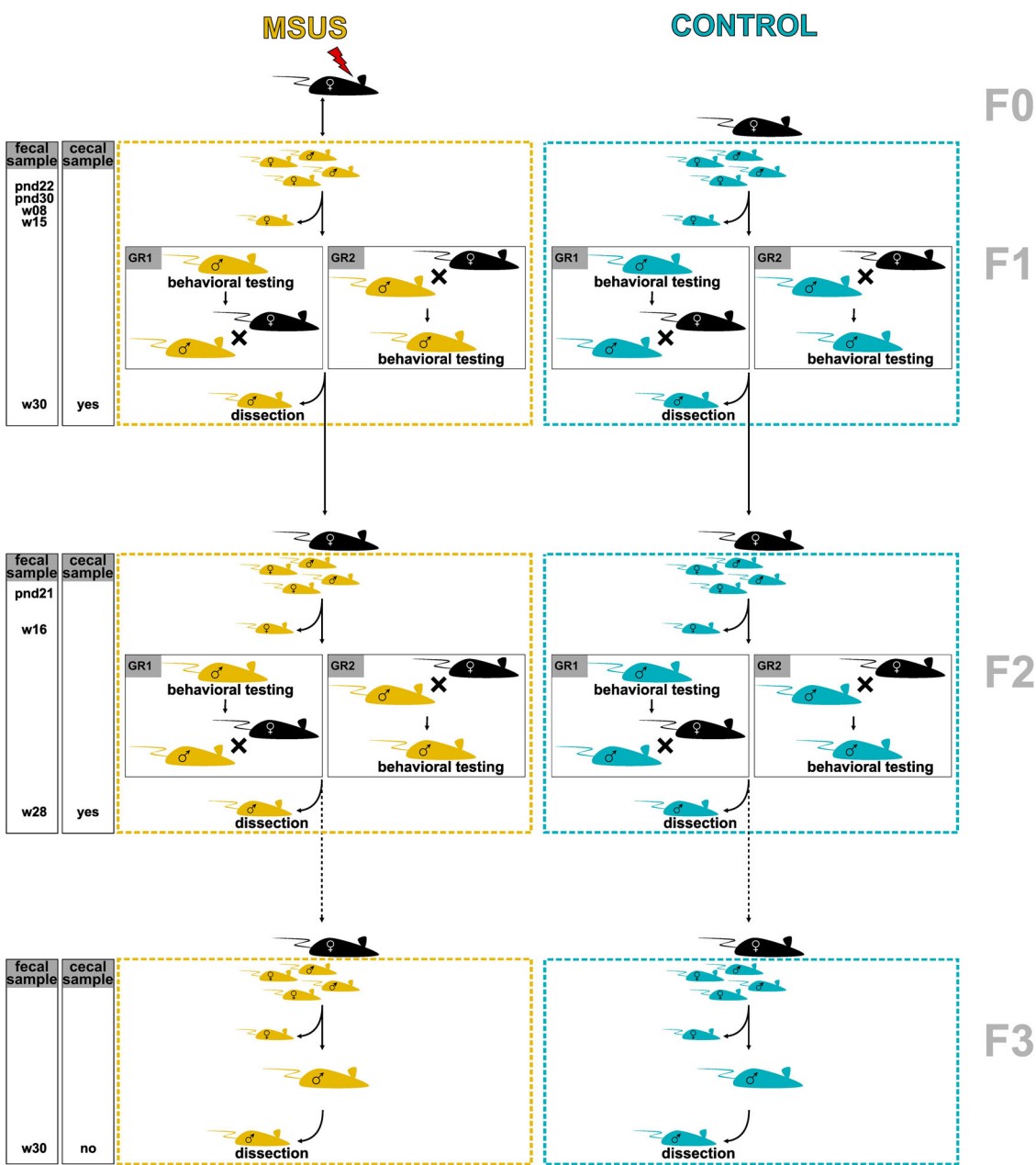

**Fig. 7 | Experimental setup of mouse model.** Setup and sample collection indicated across three generations. F3 mice are not direct offspring of displayed F2 mice (indicated by dashed line), but of a subset of F2 mice, which were not behaviorally phenotyped. F1 mice were subjected to elevated plus maze and forced swim tests and F2 mice to elevated plus maze test. Phenotyping group 1 (GR1): behavioral phenotyping before breeding. Phenotyping group 2 (GR2): breeding before behavioral phenotyping. pnd: postnatal day; w- week.

time, and addressing the coprophagous nature of rodents (i.e., cage effect) it also has limitations. Fecal samples only serve as a proxy of the gut microbial community and functionality in the gastrointestinal tract. Akin to humans, pH and the gut microbial community varies along the gastrointestinal tract of mice and is thus considerably different from fecal samples[60]. In addition, a previous study has illustrated that the murine colon mucosa-associated and lumen microbial community are differentially affected by stress[61]. Thus, observations in the present study may not be directly extrapolated to all niches of the gastrointestinal tract. Biological relevance of observed microbial changes needs to be addressed in future studies. Direct cecal bacterial metabolite quantification was limited to one timepoint each in F1 and F2. Whether changes in bacterial metabolites are present at other stages across life and following generations remains to be explored. High-resolution techniques (i.e.,

metabolomics, shotgun metagenomics, or RNA sequencing) are warranted to address potential functional changes in gut microbiota between MSUS and controls over different generations.

In conclusion, the MSUS paradigm not only changes the fecal microbial community in directly exposed mice but also in their offspring. Further well-designed studies are warranted to validate and extend on present results, using high-resolution techniques and addressing host factors responsible for gut microbial modulation. Though, our distance based analyses (based on binary and weighted Jaccard index) suggested that the fecal microbial community in F2 mice were not directly linked to the father's microbiota. Further carefully designed experiments are warranted to entirely omit potential direct transmission of gut microbes from parents to offspring. For instance, assisted reproductive techniques such as artificial insemination could be used to avoid interactions

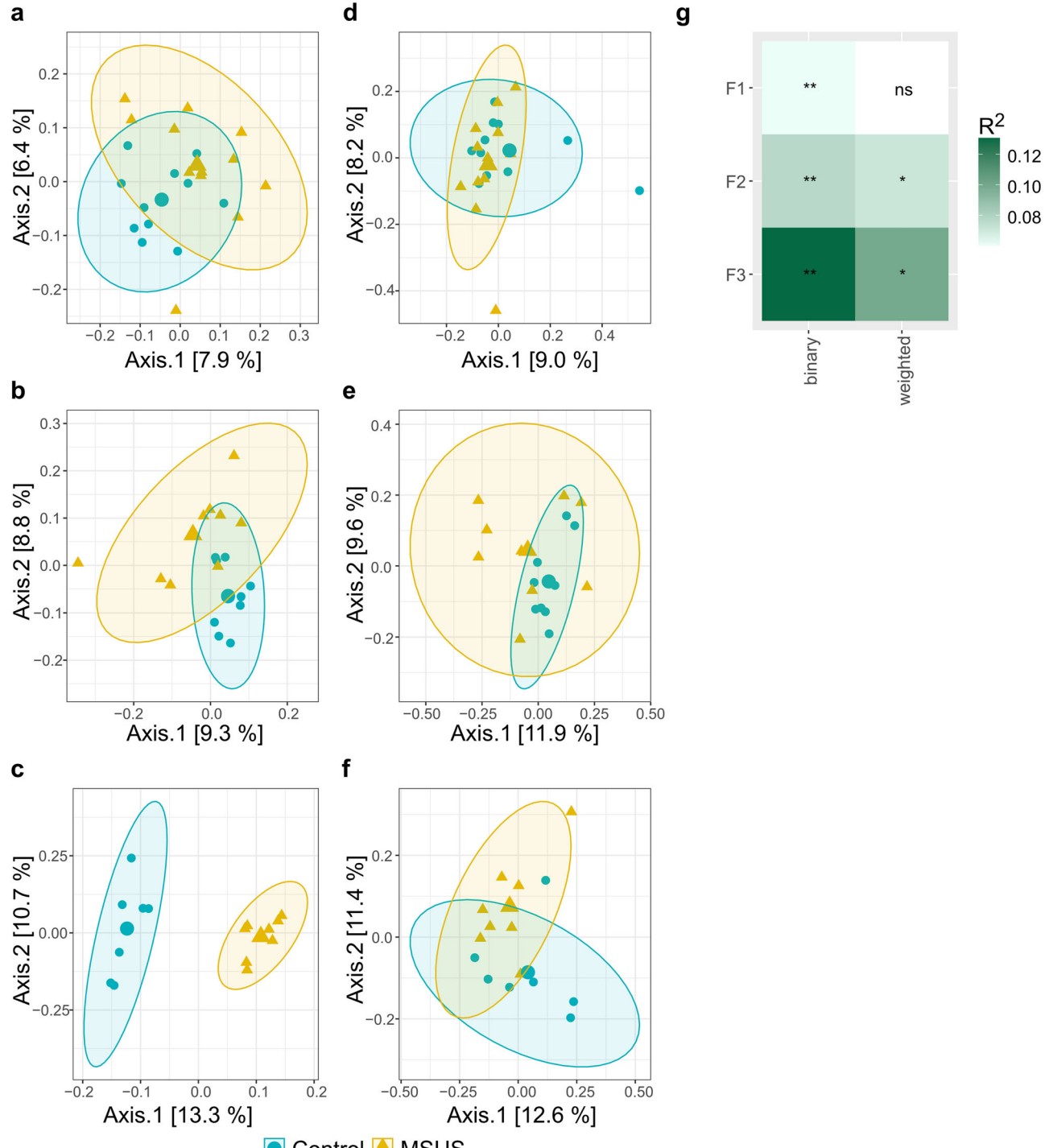

Control · MSUS

**Fig. 8 | Effect of the MSUS paradigm on beta diversity metrics of F1, F2, and F3 mice microbiota.** Comparison of MSUS *versus* control mice. Visualization as principal correspondence analysis of (**a**−**c**) binary (microbial composition) and (**d**−**f**) weighted Jaccard index (microbial structure). Small symbols display aggregated microbiota per cage, large symbols display centroids, and ellipses indicate 95% of confidence intervals. Microbial beta diversity of 30-week-old mice is depicted for (**a**, **d**) F1 and (**c**, **f**) F3, and microbial beta diversity of 28-week-old mice is depicted for (**b**, **e**) F2. **g** Heatmap of $R^2$ for comparison between MSUS and control across different generations. Significance was calculated using PERMANOVA including FDR correction. Number of animals per group are stated in Supplementary Table 4. Comparisons of different dispersions between MSUS and controls at different time points are listed in Supplementary Table 3. ns: not significant; *$p < 0.05$; **$p < 0.01$.

between parents and eliminate related social confounds[62]. Previous research has shown that early life stress modulates the murine gut microbial community in a sex-dependent manner[53]. Whether changes in gut microbial features observed in the present study are also detectable in matrilines remains to be explored.

## Material and methods
### Mouse husbandry and MSUS paradigm
We have complied with all relevant ethical regulations for animal use. Ethical approval was given by the Swiss cantonal regulations for animal experimentation under license numbers ZH057/15 and ZH083/18. All

C57Bl/6J mice were kept in a temperature- and humidity-controlled facility under a reverse 12 h light/dark cycle with access to food and water *ad libitum*. Cage change took place once a week. First generation (F1) control and MSUS mice were obtained by breeding 3-month-old C57Bl/6J

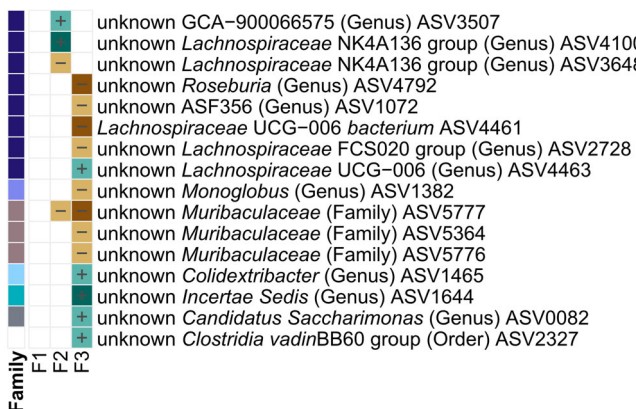

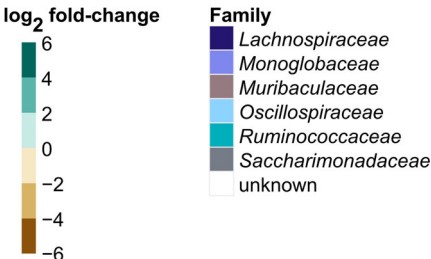

**Fig. 9 | Log2 fold-change in specific ASVs significantly (*p* < 0.05, FDR-adjusted) decreased or increased in MSUS compared to controls in F1, F2, and F3.** Differentially abundant ASVs of 30-week-old mice are depicted for F1 and F3, and differentially abundant ASVs of 28-week-old mice are depicted for F2. Taxonomic information is indicated at family level. Relative abundances of individual ASVs are depicted in Supplementary Figs. 12, 13. Significance was calculated using log₂ transformed abundance counts and generalized mixed effect models with FDR correction. Number of animals per group are stated in Supplementary Table 4.

primiparous females with age-matched males. Pairs were kept in the same cage for one week. After birth, litters/dams were randomly assigned to control and MSUS groups. For MSUS group, dams were unpredictably separated from their pups for 3 h per day from postnatal day (pnd) 1 to 14, during which each mother was randomly exposed to restrain stress (20 min restraint in a tube) or acute swim stress (5 min in cold water, 18 °C). Controls were left undisturbed. At weaning (pnd 21) male pups were randomly assigned to cages accounting for treatment and only including mice from different dams per cage, resulting in 3−5 mice per cage. After cage assignment, mice were tagged to allow for identification of individual mice throughout life. F1 control and MSUS mice were bred to naïve primiparous females to obtain second generation (F2) mice. Akin, F2 control and MSUS mice with no prior behavior testing were bred to naïve primiparous females to obtain third generation (F3) mice (Fig. 7). Mice of F3 were not bred.

The present study was part of a bigger experiment, where behavior (i.e., risk-taking behavior and behavioral despair) was evaluated *via* elevated plus maze and forced swim test as described previously [24]. To evaluate effect of temporal succession of behavioral phenotyping and breeding, after fecal collection cages with 15-week-old (F1) and 16-week-old (F2) mice were randomly assigned to group 1 (GR1; behavioral phenotyping before breeding) or group 2 (GR2; breeding before behavioral phenotyping; Fig. 7).

**Assessment of animal weight and collection of fecal and cecal samples**

Animal weight was assessed for F1 mice at 8, 15, and 30 weeks of age, and for F2 and F3 mice at 28 and 30 weeks of age, respectively. For fecal sample collection, individual mice were placed on sterilized gloves worn by animal caretaker and let to defecate. At day of sacrifice, fecal samples were directly collected from colon. For F1 mice fecal samples were collected for 22-day-old (after weaning), 30-day-old (after tagging), 8-week-old, 15-week-old, and 30-week-old (at day of sacrifice) mice (Supplementary Table 4). For F2 mice fecal samples were collected for 21-day-old (after weaning), 16-week-old, and 28-week-old (at day of sacrifice) mice (Supplementary Table 4). For both generations, cecal samples were collected at day of sacrifice. Mice were sacrificed approximately two months after behavioral testing. For F3, fecal samples were collected for 30-week-old (at day of sacrifice) mice only (Supplementary Table 4). Cecal samples of F2 dams (control: *n* = 4; MSUS: *n* = 6) were collected at day of sacrifice. All samples were frozen in liquid

**Fig. 10 | Comparison of different beta diversity metrics (binary and weighted Jaccard index) between F2 pups and their parents, and between mating pairs and non-mating pairs. a** Distance between mother or father and respective pup microbiota at pup age pnd21 and week16. **b** Distance between mating pairs (*n* = 4) and non-mating pairs (*n* = 28). Boxplot with box elements showing upper and lower quantile and median. Whiskers extending from the hinge to +/-1.5 times the interquartile range or the highest/lowest value. Colored points display individual distances and black points indicate outliers. Phenotyping group of fathers is indicated for mating pairs. Significance was calculated using Wilcoxon rank-sum test including FDR correction. pnd: postnatal day; ns: not significant; *p* < 0.05; ** *p* < 0.01; ***p* < 0.001; ****p* < 0.0001; GR1: behavioral phenotyping before breeding; GR2: breeding before behavioral phenotyping.

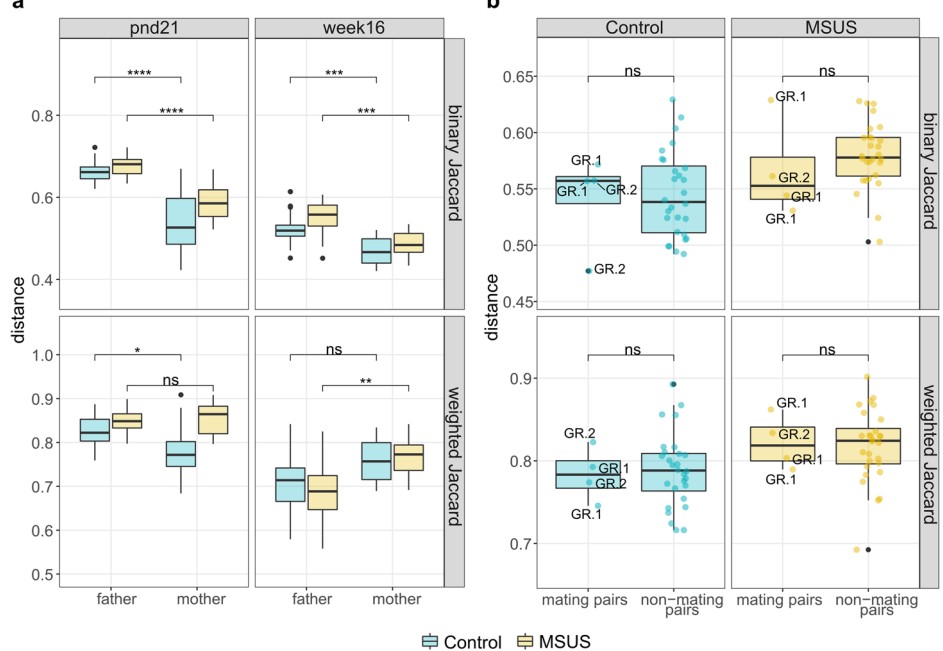

nitrogen immediately after collection and stored at −80 °C until analysis. Experimenters were blind to groups when collecting biological samples.

## Cecal bacterial metabolite quantification

To determine bacterial metabolites in cecum content, samples were homogenized with 100 mM $HClO_4$ at a 1:3 ratio (w/v) followed by two centrifugation steps (6000 x g, 20 min and 14,000 × g, 15 min, 4 °C). Resulting supernatant samples were stored at −80 °C until analysis. Samples were passed through a 0.2 µm nylon membrane prior to analysis with liquid chromatography as described previously[63]. Separation for organic acid quantification (i.e., SCFAs and intermediate metabolites) was conducted on a LaChrom HPLC-System (Merck-Hitachi, Japan) using a SecurityGuard Cartridges Carbo-H (4 × 3.0 mm; Phenomenex Inc., Torrance, CA, United States) connected to a Rezex ROA-Organic Acid H+ (300 × 7.8 mm; Phenomenex Inc.) column. Injection volume was 40 µl and elution was carried out at 40 °C under isocratic conditions, using 10 mM $H_2SO_4$ as eluent and a flow rate of 0.4 ml/min. Separated compounds were quantified using a refractive index detector L-2490 (Merck Hitachi). Raw data was analyzed using the EZChrom software (Agilent, Santa Clara, CA, United States).

For quantification of cecal amine, amino acid, and ammonia a precolumn derivatization was performed. In short, 100 µl of supernatant sample were mixed with 175 µl borate buffer (1 M $H_3BO_3$ adjusted to pH 9 with NaOH), 75 µl methanol, 4 µl internal standard (2 g/l L-2-aminoadipic acid, Sigma-Aldrich Chemie GmbH, Buchs, Switzerland), and 3.5 µl diethyl ethoxymethylenemalonate (VWR International AG, Dietikon, Switzerland) and incubated at room temperature in an ultrasonic bath for 45 min. Subsequently, samples were heated at 70 °C for 2 h to stop the derivatization reaction and passed through a 0.2 µm nylon membrane filter. Separation was carried out on an ACQUITY UPLC H-Class System (Waters Corp., Milford, MA, United States) using an ACQUITY BEH C18 column (1.7 µm particle size; 2.1 × 100 mm; Waters Corp.). Injection volume was 1 µl and elution was carried out at 40 °C with a flow rate of 0.46 ml/min, applying a gradient of 25 mM acetate buffer (pH 6.6), 100% methanol, and 100% acetonitrile as described previously[63]. Separated compounds were quantified using a diode array detector at 280 nm. Raw data was analyzed using the Empower 2 software (Waters Corp.).

## Metabarcoding of bacterial community

DNA from fecal or cecal samples was extracted using the FastDNA Spin kit for soil (MP Biomedicals, Illkirch, France) according to manufacturer's instructions. The V4 16S rRNA gene region was amplified using the primers 515 F (5′-GTGCCAGCMGCCGCGGTAA-3′) and 806 R (5′-GGAC-TACHVGGGTWTCTAAT-3′). Subsequently, a tag-encoded MiSeq-based (Illumina, CA, USA) high throughput sequencing was performed, using an Illumina MiSeq System v2 including a flow cell with 2 × 250-bp paired-end Nextera chemistry supplemented with 10% (v/v) of PhiX as sequencing control. Samples from F1 were randomized in three different sequencing runs. Samples from F2 and F3 were sequenced in two separate runs.

All raw Illumina sequencing data was processed using the R package metabaRpipe[64]. In short, adaptors and V4 primers were removed using Atropos[65] and ASVs were constructed using the DADA2 pipeline[66]. Taxonomic assignment was performed using DADA2 formatted SILVA reference base (v138.1) with confidence threshold set to 50%. Raw Illumina sequences were deposited on the European Nucleotide Archive with the accession number PRJEB57336.

## Statistics and reproducibility

Data and statistical analysis was carried out in R (v4.2.0)[67] using the packages, vegan (v2.5.7)[68], ape (v5.5)[69], ampvis2 (v2.7.9)[70], speedyseq (v0.5.3.9018)[71], Maaslin2 (v1.10.0)[72], and DivComAnalyses (v0.9)[73]. To calculate significance between two groups without confounders a Wilcoxon rank-sum test was performed. To calculate significance between groups and controlling for confounders generalized mixed effect models with false discovery rate (FDR) correction, using the Benjamini and Hochberg method, were applied. When applicable, cage, litter and sequencing run were treated as random effects

while treatment (MSUS *versus* control), age, and phenotyping group were treated as fixed effects. Models were applied *via* Microbiome Multivariable Associations with Linear Models (MaAsLin 2) with minimal prevalence set to 0.25. Number of animals per group are listed in Supplementary Table 4.

For alpha and beta diversity analyses sequences were rarified to equal sequencing depth. For comparison of alpha diversity indices generalized mixed effect models were applied *via* MaAsLin 2 using $\log_2$ transformed values. For beta diversity a linear model was fitted to various distance metrics. Homogeneity was investigated using a permutation test and effect of individual fixed effects was investigated *via* permutational multivariate analysis of variance (PERMANOVA). As random effects cannot be accounted for in PERMANOVA, an aggregation of samples per cage was performed prior to analysis to account for cage and litter effect (prior randomization of litter mates per cage). Differential abundance testing was performed with MaAsLin 2 using $\log_2$ transformed relative abundance counts.

Significance level was set to $p \leq 0.05$. Median values are stated including interquartile range (iqr) in brackets. In boxplots, outliers are indicated and defined as values greater than 1.5 times the iqr over the 75th percentile and values smaller than 1.5 iqr under the 25th percentile. Data visualization was performed in R using ggplot2 (v3.3.5)[74] and ComplexHeatmap (v2.13.1)[75].

## Reporting summary

Further information on research design is available in the Nature Portfolio Reporting Summary linked to this article.

## Data availability

Raw sequences and metadata generated and analyzed during the current study are available in the European Nucleotide Archive repository under accession number PRJEB57336. Metadata can also be accessed via BioSamples. Raw data on cecal bacterial metabolites are available as Supplementary Data 1.

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

## Acknowledgements

We thank Alfonso Die and Adam Krzystek for assistance with liquid chromatography analyzes. Illumina sequencing data were generated in collaboration with the Genetic Diversity Centre (GDC), ETH Zürich. This work was supported by own resources (ETH Zürich and University Children's Hospital Zürich). The Mansuy lab is funded by the University Zürich, the ETH Zürich, the Swiss National Science Foundation grant number 31003A_175742/1, the National Centre of Competence in Research (NCCR) RNA & Disease funded by the Swiss National Science Foundation (grant number 182880/Phase 2 and 205601/Phase 3), ETH grants (ETH-10 15-2 and ETH-17 13-2), European Union Horizon 2020 Research Innovation Program Grant number 848158, European Union projects FAMILY and HappyMums funded by the Swiss State Secretariat for Education, Research and Innovation (SERI), Free Novation grant from Novartis Forschungsstiftung, and the Escher Family Fund. L.K. received financial support from the Department of Health Science and Technology of ETH Zürich.

## Author contributions

C.L., I.M., C.B., N.O., and B.P. designed the study. N.O. and L.K. performed the experiments. N.O. and F.C. performed the data analysis. C.B., C.L., and I.M. provided financial support. N.O., B.P., and C.L. drafted, and all authors critically reviewed the manuscript.

## Competing interests

The authors declare no competing interests.
