## [Peer Review File · Communications Biology]

Reviewers' comments:

Reviewer #1 (Remarks to the Author):

Thank you for the chance to review “Transgenerational effects of early life stress on the fecal microbiota in mice,” an original research paper examining the relationship between early life stress and fecal microbial composition across multiple generations. The study was well done, with a key strength being its longitudinal design - assessing multiple generations. The article is very well written and will be an interesting addition to the literature, with some minor issues clarified below.

- p3: The authors mention socioeconomic risk exposure; was this childhood socioeconomic exposure or adulthood? If childhood, was childhood SES confounded with adulthood SES?
- p4: Can the authors clarify what they mean by combining stress of mothers and of pups?
- p23: When the authors say “early life stress exposure alters host factors, which modulate gut microbes,” what are some of these host factors?

Reviewer #2 (Remarks to the Author):

In this paper, Otaru et al. explore an intriguing topic by presenting data on the potential transgenerational effects of early-life stress on the fecal microbiota in mice. They demonstrate that early stress induces significant and persistent changes in fecal microbial composition and structure (Beta diversity) throughout the lifespan. Multiple amplicon sequencing variants (ASVs) exhibit differential abundance between early-stressed and control mice, and some ASVs exhibit persistent decreases in both F2 and F3 offspring descended from male mice exposed to early stress.

Overall, I believe this paper is of significant importance to the field, providing valuable insights into the transgenerational effects of early-life stress on the fecal microbiota.

I suggest strengthening the paper by incorporating or specifying certain aspects as follows:

1. Expand the introduction by providing a more detailed exploration of transgenerational data related to early-life stress.
2. While the results are intriguing and well-presented, the discussion seems somewhat superficial and unclear. I recommend a more in-depth analysis of certain results:
 - Although the primary focus of the paper does not center on the dynamics and functionality of fecal microbiota from late postnatal to adult stages, the authors might briefly comment on the results in the discussion.
 - The ASVs analysis showed a decrease in the Ruminococcaceae family in both 8-week and 15-week-old early-stressed mice. Furthermore, in F2 and F3 offspring descended from male mice exposed to early

stress, half of the differentially abundant ASVs belonged to the Lachnospiraceae family, while the ASV belonging to the Muribaculaceae family exhibited persistent decreases in both F2 and F3 offspring mice. Discussing the implications of these findings, such as the potential role of these families and providing evidence linking them to early stress, would be beneficial.

- Consider discussing the observed differences in specific ASVs and metabolic pathways between the GR1 and GR2 groups.
- Provide a clearer description of the paper by Palma and colleagues (Lines 63 to 380). Additionally, clarify the connection with microRNAs, as this aspect was not fully understood.

3. In figure 10, it would be useful to distinguish between F0 and F1 generations. Additionally, I suggest specifying the details of the behavioral testing in the caption.

4. Line 143: the result on the Beta-diversity between GR1 and GR2 is presented in the Supplementary table 1, not in the Supplementary table 2.

5. Line 374: remove “being”

Reviewer #4 (Remarks to the Author):

The study from Otaru et al entitled “Transgenerational effects of early life stress on the fecal microbiota in mice” describes data obtained with 16S rRNA amplicon sequencing, bacterial metabolic pathway prediction using the PICRUSt2 software as well as metabolite quantification of cecal contents of mice across three generations. Together, the results show that early life traumatic stress in parents shifts the fecal microbial composition and structure, and the changes persisted in the progeny across two generations. Although the changes included increases in relative abundance of bacteria belonging to the genera *Bifidobacterium* and *Lachnospiraceae* (Gr2 vs GR1) as well as *Roseburia* (F1 MSUS compared to controls over life), these effects are not accompanied by changes in short-chain fatty acids, branched-chain fatty acids or other bacterial metabolites (and related predicted metabolic pathways) in any generation.

I have the following comments that should be addressed before I feel that the manuscript will be suitable for publication:

The authors used PICRUSt2 to predict fecal metabolic pathways based on the ASVs detected in the fecal samples from mice in the different trials. Accuracy of prediction was determined via the abundance-weighted nearest sequenced taxon index (NSTI) and it is reported to have an average prediction based on taxa with 82% similarity (L130) for the control mice fecal microbiota. Average prediction for fecal metabolites based on differences between MSUS and control mice for all tested ages was based on taxa with 83% similarity (L219); in addition, the average prediction for the metabolic potential in MSUS and for F1, F2, and F3 was based on taxa with 82% similarity (282). As the authors highlight in the discussion

section fairly, bacterial metabolic pathway prediction via PICRUSt2 has several limitations, especially for non-human samples. The numbers for the accuracy of the predictions obtained in this study are indeed concerning. Thresholds of 98.7%, 94.5%, and 86.5% 16S rRNA gene sequence identities are considered as indications for novel species, genera, and families, respectively (see PMID: 25118885), leaving this reviewer wondering about the significance of the predictions given that the similarities between the mouse gut bacteria and PICRUSt2-genomes are at the family level (and way too distant to be reliable in a mouse context). Considering that authors have obtained DNA for all the samples presented in this study, shotgun metagenomics seem to be the natural way of obtaining unequivocal results that can support the authors' claims in this paper.

L147-L148; 10 predicted metabolic pathways, related to carbohydrate and amino acid metabolism, were differentially abundant between groups; however, no differences in measured cecal bacterial SCFAs, organic acids, amino acids and amines were detected. In addition, in figure 3a, it appears that the relative abundance of bifidobacteria (acetate and lactate producers) and lachnospiraceae (propionate and butyrate producers) increase. Could the author comment on these contrasting results in the manuscript, proposing a possible explanation?

L229-L230. Could the authors clarify why they used F3 mice that were not direct offspring of the investigated F2 mice, but of a subset of F2 mice which were not included in this study?

L370 – 380. This section is very speculative, and no results related to small non-coding RNA have been presented in the present study.

We would like to thank the reviewers for the constructive comments, which have significantly contributed to the improvement of the manuscript. We have carefully addressed each point raised by the reviewers, providing detailed responses and making corresponding revisions to the manuscript, as outlined below. Cited line numbers refer to the line in the corrected manuscript (track change version).

Reviewer #1:

P3: The authors mention socioeconomic risk exposure; was this childhood socioeconomic exposure or adulthood? If childhood, was childhood SES confounded with adulthood SES?

We thank the reviewer for this comment. The study we are referring to associate the economic status of parents (i.e. low income to needs ratio) and social aspects (e.g. family turmoil and parent-child dysfunction) with fecal microbial taxonomic and functional diversity of the children. Since the study was not a longitudinal study, we cannot comment on the correlation of childhood and adulthood socioeconomic risk exposure. To clarify the different factors the manuscript has been adapted as follows:

L64: “In the few available human studies referring to gut microbial metabolic changes associated with early life stress, socioeconomic risk exposure (e.g. low income to needs ratio and parent-child dysfunction) explained a significant amount of the functional gut microbial diversity (Flannery et al. 2020) and a decrease of four gut metabolites (i.e. glutamate gamma-methyl ester, 5-oxoproline, malate, and urate) was observed when comparing individuals exposed to early life adversities (Coley et al. 2021).”

p4: Can the authors clarify what they mean by combining stress of mothers and of pups?

Stress of mothers and pups is combined because both are subjected to daily unpredictable separation for 3 h and in addition mothers are stressed unpredictably during separation by restraint or forced swim. This is therefore a combined separation stress and physical stress for mothers. The corresponding paragraph now reads as follows:

L74: “The mouse model of unpredictable maternal separation combined with unpredictable maternal stress (MSUS) distinguishes itself from other rodent models of stress by applying unpredictability as traumatic factor and by combining stress of mothers and of pups (daily unpredictable separation for 3 h and in addition mothers are stressed unpredictably during separation by restraint or forced swim) (Franklin et al. 2010).”

p23: When the authors say “early life stress exposure alters host factors, which modulate gut microbes,” what are some of these host factors?

We have adapted the corresponding paragraph in the manuscript to include an example of host factors (L435). The paragraph now reads as follows:

L435: “These observations suggest that early life stress exposure modulates host factors (e.g. immune tone (Wouw et al. 2020; Xu et al. 2020)). These host factors may be responsible for modulation of gut microbes. The altered gut microbial community may then signal back to the host exhibiting phenotypic changes.”

Reviewer #2

1. Expand the introduction by providing a more detailed exploration of transgenerational data related to early-life stress.

As recommended by the reviewer, we added a short paragraph in the introduction on the question of transgenerational data related to early life stress.

L39: “Thus, accumulating evidence indicates that the effects of early life stress can be manifested by subsequent generations, potentially affecting their behavior and physiology (Jawaid et al. 2021). A first study by Weaver et al. (2004) showed that maternal care in rats may alter the stress responses of the offspring by inducing epigenetic changes, particularly changes in DNA methylation at the promoter of the glucocorticoid receptor gene in the hippocampus. This effect is mediated by maternal care and needs exposure at each generation (Weaver et al. 2004). Transmission of the effects of early life stress involving the germline was however demonstrated in mice. Male mice subjected to early life stress have behavioral and metabolic symptoms that can be manifested in their offspring until the fourth or even fifth generation (van Steenwyk et al. 2018, Boscardin et al. 2022). Evidence of cross-generation effects of early life stress in humans is scarcer and remains correlative (Zhou and Ryan 2023) but suggest a link between adversity and gut microbiome composition (Querdasi et al 2022). “

2. While the results are intriguing and well-presented, the discussion seems somewhat superficial and unclear. I recommend a more in-depth analysis of certain results:

Although the primary focus of the paper does not center on the dynamics and functionality of fecal microbiota from late postnatal to adult stages, the authors might briefly comment on the results in the discussion.

We thank and agree with the reviewer's comment. We have therefore adapted the manuscript to include a section briefly discussing age related changes in fecal microbiota of mice. The manuscript now includes the following:

L346: “During the course of life distinct gut microbial changes occur, though in humans it is difficult to distinguish between lifestyle changes and the aging process itself (Skillington et al. 2021; Salazar et al. 2020). In line with human studies (Bergström et al. 2014), here we show that the cessation of breastmilk has the most pronounced effect on the murine gut microbiota, illustrated by changes in gut microbial features including richness, evenness, structure, and composition. Although richness and evenness stay relatively stable in adulthood, significant changes are still observed over time with respect to structure and composition, suggesting a significant effect of the aging process itself. This is in line with previous studies addressing age-related changes in the murine gut microbiota, demonstrating the dynamics of the fecal microbial community and its metabolic functions from young adult (approximately 8 weeks of age) to old (older than 100 weeks of age) mice (Low et al. 2022; Langille et al. 2014; You et al. 2022).”

The ASVs analysis showed a decrease in the Ruminococcaceae family in both 8-week and 15-week-old early-stressed mice. Furthermore, in F2 and F3 offspring descended from male mice exposed to early stress, half of the differentially abundant ASVs belonged to the Lachnospiraceae family, while the ASV belonging to the Muribaculaceae family exhibited persistent decreases in both F2 and F3 offspring mice. Discussing the implications of these findings, such as the potential role of these families and providing evidence linking them to early stress, would be beneficial.

We thank the reviewer for this helpful comment. In response, we have included corresponding literature and have adapted the manuscript to include the following paragraphs:

L372: “We show that one ASV belonging to the family *Ruminococcaceae* is persistently decreased in both 8 and 15-week-old mice, in line with a previous study, illustrating the depletion of corresponding ASVs after early life stress exposure (Kemp et al. 2021). The *Ruminococcaceae* family includes prominent butyrate producers (Louis and Flint, 2017), suggesting a modulation of the SCFA-profile by early life stress. Yet, here we report no significant changes in cecal SCFAs, amines, amino acids between early life stress exposed and non-exposed 30-week-old mice.”

L415: “However, only one ASV belonging to the family *Muribaculaceae* was persistently decreased in MSUS compared to controls in F2 and F3, suggesting different shifts in fecal bacterial community in MSUS mice of different generations. Yet, different ASVs belonging to the family *Muribaculaceae* were depleted in both F2 and F3. The *Muribaculaceae* abundance has previously been correlated with T-cells (Francella et al. 2022), depressive-like (Li et al. 2021), and autism-like behavior (Kong et al. 2021). In line with directly early life stress exposed mice (Kemp et al. 2021; Park et al. 2021; Reemst et al. 2022), we show a modulation of *Lachnospiraceae* ASVs in both F2 and F3. Abundance of the family *Lachnospiraceae* has previously been negatively correlated with depressive-like behavior in mice (Guida et al. 20218). The family *Lachnospiraceae* harbors major butyrate and propionate producers (Louis and Flint, 2017), while *Muribaculaceae* have previously been associated with propionate production (Smith et al. 2019). This may imply a modulation of SCFA levels in F2 and F3, which we could not confirm here. This discrepancy could be explained by different factors, including absorption of SCFAs in the cecum, time of feeding before sacrifice, and the concept of functional redundancy (Moya et al. 2016; Besten et al. 2013). Nevertheless, these observations suggest a crucial role of both *Muribaculaceae* and *Lachnospiraceae* species in the microbiota-host crosstalk.”

Consider discussing the observed differences in specific ASVs and metabolic pathways between the GR1 and GR2 groups.

The discussion was modified to include a corresponding discussion section:

L355: “We show that the temporal succession of breeding and behavioral phenotyping of mice has a significant effect on relative abundances of several fecal microbial ASVs. Both acute and chronic stress in adulthood have been shown to alter the fecal and cecal microbial community and related metabolic functions in rodents (Qu et al. 2021; Zhang et al. 2021; Kim et al. 2019; Bassett et al. 2019; Rao et al.2021; Seewoo et al. 2022; Bridgewater et al. 2017). As the applied behavior phenotyping methods, i.e. forced swim test, pose as an acute stressor themselves, it is conceivable that observed alterations originate in the temporal differences between acute stressor and fecal sample collection. However, fecal sample collection for both groups (behavioral phenotyping before breeding *versus* breeding before behavior phenotyping) was performed at least two months after behavioral phenotyping. Whether two months are indeed sufficient to alleviate acute stress effects on the gut microbiota remains to be explored. “

Provide a clearer description of the paper by Palma and colleagues (Lines 363 to 380). Additionally, clarify the connection with microRNAs, as this aspect was not fully understood.

We provided a clearer description of the paper of Palma and colleagues (L429). However, we chose to remove statements on microRNAs (L440-L451) because we did not present corresponding data related to small non-coding RNAs and therefore we felt that this connection was indeed speculative in the presented study. The paragraph was restructured as follows:

L429: “Similarly, experiments with germ free mice have suggested a crucial involvement of gut microbes in anxiety-like behavior and behavioral despair induced by early life stress (Palma et al. 2015). After colonization of early life stressed germ-free mice distinct shifts in gut

microbiota profiles were observed, which were not present in unstressed controls. It was only with said colonization that differences in anxiety-like behavior and behavioral despair between early life stressed mice and unstressed controls were detected (Palma et al. 2015).“

3. In figure 10, it would be useful to distinguish between F0 and F1 generations. Additionally, I suggest specifying the details of the behavioral testing in the caption.

As suggested by the reviewer, we have updated Figure 10 indicating F0 and F1 (see figure below). Additionally, we have added further details regarding behavioral testing in the caption of Figure 10:

L515: “F1 mice were subjected to elevated plus maze and forced swim tests and F2 mice to elevated plus maze test.”

4. Line 143: the result on the Beta-diversity between GR1 and GR2 is presented in the Supplementary table 1, not in the Supplementary table 2.

We apologize for the typo mistake and corrected accordingly (L159)

5. Line 374: remove “being”

As indicated above we chose to remove statements on microRNAs (L440-L451) which included the section to be corrected.

Reviewer #4:

The authors used PICRUSt2 to predict fecal metabolic pathways based on the ASVs detected in the fecal samples from mice in the different trials. Accuracy of prediction was determined via the abundance-weighted nearest sequenced taxon index (NSTI) and it is reported to have an average prediction based on taxa with 82% similarity (L130) for the control mice fecal microbiota. Average prediction for fecal metabolites based on differences between MSUS and control mice for all tested ages was based on taxa with 83% similarity (L219); in addition, the average prediction for the metabolic potential in MSUS and for F1, F2, and F3 was based on taxa with 82% similarity (282). As the authors highlight in the discussion section fairly, bacterial metabolic pathway prediction via PICRUSt2 has several limitations, especially for non-human samples. The numbers for the accuracy of the predictions obtained in this study are indeed concerning. Thresholds of 98.7%, 94.5%, and 86.5% 16S rRNA gene sequence identities are considered as indications for novel species, genera, and families, respectively (see PMID: 25118885), leaving this reviewer wondering about the significance of the predictions given that the similarities between the mouse gut bacteria and PICRUSt2-genomes are at the family level (and way too distant to be reliable in a mouse context). Considering that authors have obtained DNA for all the samples presented in this study, shotgun metagenomics seem to be the natural way of obtaining unequivocal results that can support the authors' claims in this paper.

We thank the reviewer for the helpful comment. We fully agree that PICRUSt2 analyses have important limits as presented in the comment. We also agree that metagenomics analyses could be a good alternative when this would have been possible. Thus, we chose to exclude the corresponding sections on PICRUSt2 prediction of metabolic potential. We believe that this

omission is not impacting the scientific message of our study. The entire manuscript has been adjusted accordingly.

L147-L148; 10 predicted metabolic pathways, related to carbohydrate and amino acid metabolism, were differentially abundant between groups; however, no differences in measured cecal bacterial SCFAs, organic acids, amino acids and amines were detected. In addition, in figure 3a, it appears that the relative abundance of bifidobacteria (acetate and lactate producers) and Lachnospiraceae (propionate and butyrate producers) increase. Could the author comment on these contrasting results in the manuscript, proposing a possible explanation?

As indicated above, we chose to remove the PICRUST2 predictions of metabolic potential. The inconsistency of observed compositional changes with no accompanying change in SCFA levels could be explained by different factors which we have included in the manuscript as follows:

L423: “The family *Lachnospiraceae* harbors major butyrate and propionate producers (Louis and Flint, 2017), while *Muribaculaceae* have previously been associated with propionate production (Smith et al. 2019). This may imply a modulation of SCFA levels in F2 and F3, which we could not confirm here. This discrepancy could be explained by different factors, including absorption of SCFAs in the cecum, time of feeding before sacrifice, and the concept of functional redundancy (Moya et al. 2016; Besten et al. 2013).”

L229-L230. Could the authors clarify why they used F3 mice that were not direct offspring of the investigated F2 mice, but of a subset of F2 mice which were not included in this study?

This was for practical reasons. We could not use the direct F3 descendants of F2 animals because these animals were used for other analyses requiring experimental procedures not compatible with microbiome analyses and that may have biased results (e.g. extensive behavioral testing).

L370 – 380. This section is very speculative, and no results related to small non-coding RNA have been presented in the present study.

We fully agree with the reviewer's comment. The corresponding section has been shortened and statements relating to small non-coding RNA have been omitted (L440-L451)

REVIEWERS' COMMENTS:

Reviewer #1 (Remarks to the Author):

Thank you for the thorough response to the reviewers. Only a few minor comments:

-Line 69: what is the comparison group? Should this say in comparison to controls?

-Line 366 – 367: Were these human or rodent studies? Please specify in text.

- Was citation 17 meant to be Hantsoo L, Jašarević E, Criniti S, McGeehan B, Tanes C, Sammel MD, Elovitz MA, Compher C, Wu G, Epperson CN. Childhood adversity impact on gut microbiota and inflammatory response to stress during pregnancy. *Brain Behav Immun*. 2019 Jan;75:240-250. doi: 10.1016/j.bbi.2018.11.005. Epub 2018 Nov 3. PMID: 30399404; PMCID: PMC6349044. ?

Reviewer #2 (Remarks to the Author):

The revised version of the manuscript adequately addresses my queries. Therefore, I have no further remarks, and I believe that the paper is now suitable for publication.

Reviewer #4 (Remarks to the Author):

The authors have sufficiently addressed all of my previous concerns.

We would like to thank the reviewer for the additional comments. We have carefully addressed each point raised by the reviewer, providing detailed responses and making corresponding revisions to the manuscript, as outlined below. Cited line numbers refer to the line in the corrected manuscript (clean version).

Reviewer #1:

Line 69: what is the comparison group? Should this say in comparison to controls?

We thank the reviewer for this comment and apologies for the confusing wording. In the described study individuals with low and high Early Traumatic Inventory-Self Report scores were compared. To clarify the comparison, the sentence now reads as follows:

L65: “In the few available human studies referring to gut microbial metabolic changes associated with early life stress, socioeconomic risk exposure (e.g. low income to needs ratio and parent-child dysfunction) explained a significant amount of the functional gut microbial diversity (Flannery et al. 2020) and a decrease of four gut metabolites (i.e. glutamate gamma-methyl ester, 5-oxoproline, malate, and urate) was observed when comparing individuals with low and high early life adversity exposure (Coley et al. 2021).”

Line 366 – 367: Were these human or rodent studies? Please specify in text.

We have adapted the corresponding sentence in the manuscript to further specify that we refer to mouse studies. The sentence now reads as follows:

L231: “Early life stress has previously been shown to alter the murine fecal microbial community in studies focusing on a single time point after exposure (Kemp et al. 2021; Usui et al. 2021; Reemst et al. 2022)”

Was citation 17 meant to be Hantsoo L, Jašarević E, Criniti S, McGeehan B, Tanes C, Sammel MD, Elovitz MA, Compher C, Wu G, Epperson CN. Childhood adversity impact on gut microbiota and inflammatory response to stress during pregnancy. Brain Behav Immun. 2019 Jan;75:240-250. doi: 10.1016/j.bbi.2018.11.005. Epub 2018 Nov 3. PMID: 30399404; PMCID: PMC6349044. ?

With citation 17 we are referring to a review paper by Hantsoo, L. & Zemel, B. S. entitled “Stress gets into the belly: Early life stress and the gut microbiome.”, published in 2021. In this publication Hantsoo and Zemel are reviewing data on gut microbial compositional changes induced by different stresses, including both animal and human data. We think this publication nicely illustrates the heterogenous results observed in previous studies, which we refer to in our manuscript. We agree with the reviewer, that the publication of Hantsoo et al., 2019 also provides intriguing data, further exploring the role of dietary interventions to ameliorate effects of early life stress. As this publication is included in the review of 2021 we refrained from including it in our reference list.